# From BLEU to RAINBOW: Why We Need New Metrics for NLG.

## Abstract

The majority of NLG evaluation relies on automatic metrics, such as BLEU. In this paper, we investigate a wide range of these metrics, including state-of-the-art word-based and novel grammar-based ones, and demonstrate that they only weakly reflect human judgements of system outputs as generated by data-driven, end-to-end NLG. A detailed error analysis shows that automatic metrics are particularly bad in distinguishing outputs of medium and good quality, which can be partially attributed to the fact that human judgements and metrics are given on different scales. We also show that metric performance is data- and system-specific. We then suggest an alternative metric, called RAINBOW, combining the individual strengths of different automatic scores. This new metric achieves up to $\rho = 0.81$ correlation with human judgements at the sentence-level (compared to a maximum of $\rho = 0.33$ for existing metrics) and achieves stable results across systems and datasets.

## 1 Introduction

Automatic evaluation measures, such as BLEU (Papineni et al., 2002), are used with increasing frequency to evaluate Natural Language Generation (NLG) systems: Up to 60% of NLG research published between 2012–2015 relies on automatic metrics (Gkatzia and Mahamood, 2015). Automatic evaluation is popular because it is cheaper and faster to run than human evaluation, and is needed for automatic benchmarking and tuning of algorithms. The use of such metrics is, however, only sensible if they are known to be sufficiently correlated with human preferences. This is rarely

the case, as shown by various studies in NLG (Reiter and Belz, 2009; Belz and Reiter, 2006; Stent et al., 2005), as well as in related fields, such as dialogue systems (Liu et al., 2016), machine translation (MT), e.g. (Callison-Burch et al., 2006), and image captioning, e.g. (Elliott and Keller, 2014; Kilickaya et al., 2017). This paper follows on from this work and presents another evaluation study into automatic metrics with the aim to firmly establish the need for new metrics. We also suggest an alternative metric, which we call RAINBOW to reflect the diverse set of features it is based upon. In contrast to previous work, we are the first to:

• Target end-to-end data-driven NLG, where we compare 3 different approaches. In contrast to NLG methods evaluated in previous work, our systems can produce ungrammatical output by (1) generating word-by-word, and (2) learning from noisy data.

• Compare a large number of 21 automated metrics, including grammar-based ones.

• Report results on two different domains and three different datasets, which allows us to draw more general conclusions.

• Suggest an alternative automatic metric, which shows high correlation with human judgements.

• Conduct a detailed error analysis, which suggests that, while metrics can be reasonable indicators at the system-level, they are not reliable at the sentence-level.

• Make all associated code and data publicly available, including detailed analysis results.

## 2 End-to-End NLG Systems

In this paper, we focus on recent end-to-end, data-driven NLG methods, which jointly learn sentence planning and surface realisation from non-aligned data (Dušek and Jurčíček, 2015; Wen et al., 2015; Mei et al., 2016; Wen et al., 2016; Sharma et al.,

2016; Dušek and Jurčíček, 2016; Lampouras and Vlachos, 2016). These approaches do not require costly semantic alignment between Meaning Representations (MR) and human references (also referred to as "ground truth" or "targets"), but are based on parallel datasets, which can be collected in sufficient quality and quantity using effective crowd-sourcing techniques, e.g. (Novikova et al., 2016), and as such, enable rapid development of NLG components in new domains. In particular, we compare the performance of the following systems:

• **RNNLG:**[1] The system by Wen et al. (2015) uses a Long Short-term Memory (LSTM) network to jointly address sentence planning and surface realisation. It augments each LSTM cell with a gate that conditions it on the input MR, which allows it to keep track of MR contents generated so far.

• **TGEN:**[2] The system by Dušek and Jurčíček (2015) learns to incrementally generate deep-syntax dependency trees of candidate sentence plans (i.e., which MR elements to mention and the overall sentence structure). Surface realisation is performed using a separate, domain-independent rule-based module.

• **LOLS:**[3] The system by Lampouras and Vlachos (2016) learns sentence planning and surface realisation using Locally Optimal Learning to Search (LOLS), an imitation learning framework which learns using BLEU and ROUGE as non-decomposable loss functions.

| sys/ data | SFHOTEL | SFREST | BAGEL |
|---|---|---|---|
| RNNLG | ✓ | ✓ | |
| TGEN | | | ✓ |
| LOLS | ✓ | ✓ | ✓ |

Table 1: Systems and datasets used in this study.

## 3 Datasets

We consider the following datasets collected via crowd-sourcing, which target utterance generation for spoken dialogue systems. Table 1 shows which NLG system was trained on which dataset. Each instance consists of one MR and one or more natural language references as produced by humans, such as the following example, taken from the BAGEL dataset. Note that we use lexicalised ver-

sions of SFHOTEL and SFREST and a partially lexicalised version of BAGEL, where proper names and place names are replaced by placeholders (X), in correspondence with the outputs generated by the systems, as provided by the system authors.

> **MR:** inform(name=X, area=X, pricerange=moderate, type=restaurant)
> Reference: "*X is a moderately priced restaurant in X.*"

• **SFHOTEL & SFREST** (Wen et al., 2015) provide information about hotels and restaurants in San Francisco. There are 8 system dialogue act types, such as *inform, confirm, goodbye* etc. Each domain contains 12 attributes, where some are common to both domains, such as *name, type, pricerange, address, area,* etc., and the others are domain-specific, e.g. *food* and *kids-allowed* for restaurants; *hasinternet* and *dogs-allowed* for hotels. For each domain, around 5K human references were collected with 2.3K unique human utterances for SFHOTEL and 1.6K for SFREST. The number of unique system outputs produced is 1181 for SFREST and 875 for SFHOTEL.

• **BAGEL** (Mairesse et al., 2010) provides information about restaurants in Cambridge. The dataset contains 202 aligned pairs of MRs and 2 corresponding references each. The domain is a subset of SFREST, including only the *inform* act and 8 attributes.

## 4 Metrics

### 4.1 Word-based metrics (WBMs)

NLG evaluation has borrowed a number of automatic metrics from related fields, such as MT, summarisation, or image captioning, which compare output texts generated by systems to ground-truth references produced by humans. We refer to this group as word-based metrics. In general, the higher these scores are, the better or more similar to the human references the output is.[4] The following order reflects the degree these metrics move from simple $n$-gram overlap to also considering term frequency (TF-IDF) weighting and semantically similar words.

• **Word-overlap metrics:** We consider frequently used metrics, including TER (Snover et al., 2006), BLEU (Papineni et al., 2002), ROUGE (Lin, 2004), NIST (Doddington, 2002), LEPOR (Han et al., 2012), CIDEr (Vedantam et al., 2015), and METEOR (Lavie and Agarwal, 2007).

---

[1] https://github.com/shawnwun/RNNLG
[2] https://github.com/UFAL-DSG/tgen
[3] https://github.com/glampouras/JLOLS_NLG

---

[4] Except for TER whose scale is reversed.

• **Semantic Similarity (SIM):** We calculate the Semantic Text Similarity measure provided by Han et al. (2013). This measure is based on distributional similarity and Latent Semantic Analysis (LSA) and is further complemented with semantic relations extracted from WordNet.

### 4.2 Grammar-based metrics (GBMs)

Grammar-based measures have been explored in related fields, such as MT (Giménez and Màrquez, 2008) or grammatical error correction (Napoles et al., 2016), and, in contrast to WBMs, do not rely on ground-truth references. To our knowledge, we are the first to consider GBMs for sentence-level NLG evaluation.

• **Readability** quantifies the difficulty with which a reader understands a text, as used for e.g. evaluating summarisation (Kan et al., 2001) or text simplification (Francois and Bernhard, 2014). We measure readability by the Flesch Reading Ease score (RE) (Flesch, 1979), which calculates a ratio between the number of characters per sentence, the number of words per sentence, and the number of syllables per word. Higher readability score indicates a less complex utterance that is easier to read. We also consider related measures, such as characters per utterance (**len**) and per word (**cpw**), words per sentence (**wps**), syllables per sentence (**sps**) and per word (**spw**), as well as polysyllabic words per utterance (**pol**) and per word (**ppw**). The higher these scores, the more complex the utterance.

• **Grammaticality:** In contrast to previous NLG methods, our corpus-based systems can produce ungrammatical output by (1) generating word-by-word, and (2) learning from noisy data. As a first approximation of grammaticality, we measure the parsing score (**prs**) as returned by the Stanford parser, as well as the number of misspellings (**msp**). The lower these scores are, the more grammatically correct an utterance is. Note that the Stanford parser score is not designed to measure grammaticality, however, it will generally prefer a grammatical parse to a non-grammatical one.[5] In future work, we aim to use specifically designed grammar-scoring functions, e.g. (Napoles et al., 2016), once they become publicly available.

---

[5] http://nlp.stanford.edu/software/parser-faq.shtml

## 5 Human Data Collection

To collect human rankings, we presented the MR together with 2 utterances generated by different systems side-by-side to crowdworkers, which were asked to score each utterance on a 6-point Likert scale for:

• **Informativeness:** *Does the utterance provide all the useful information from the meaning representation?*

• **Naturalness:** *Could the utterance have been produced by a native speaker?*

• **Quality:** *How do you judge the overall quality of the utterance in terms of its grammatical correctness and fluency?*

Each system output was scored by 3 different crowdworkers. To reduce participants' bias, the order of appearance of utterances produced by each system was randomised and crowdworkers were restricted to evaluate a maximum of 20 utterances. The crowdworkers were selected from English-speaking countries only, based on their IP-addresses, and asked to confirm that English was their native language.

To assess the reliability of ratings, we calculated the intra-class correlation coefficient (ICC), which measures inter-observer reliability on ordinal data for more than two raters (Landis and Koch, 1977). The overall ICC across all three datasets is 0.45 ($p$ <0.001), which corresponds to a moderate agreement. In general, we find consistent differences in inter-annotator agreement per system and dataset, with lower agreements in LOLS than in RNNLG and TGEN. Agreement is highest for the SFHOTEL dataset, followed by SFREST and BAGEL (for details see Appendix A, Table 7).

## 6 System Evaluation

Table 2 summarises the individual systems' performance in terms of automatic and human scores.[6] All WBMs produce the same (significant) results, whereas GBMs show the same trend, but with different levels of statistical significance, with only **len**, **wps** and **sps** producing reliable significant results. System performance is dataset-specific: For WBMs, the LOLS system consistently produces better results on BAGEL compared to TGEN, while for SFREST and SFHOTEL, LOLS is outperformed by RNNLG with WBMs. We observe that human *informativeness* ratings follow

---

[6] Detailed results are submitted as supplementary material. See Appendix A, Table 8.

| metric | BAGEL | | SFHOTEL | | SFREST | |
|---|---|---|---|---|---|---|
| | TGEN | LOLS | RNNLG | LOLS | RNNLG | LOLS |
| WBMs | | More overlap | More overlap* | | More overlap* | |
| SIM | More similar | | More similar* | | | More similar |
| RE | | More complex(*) | | More complex(*) | | More complex(*) |
| GBMs | Better grammar(*) | | Better grammar(*) | | Better grammar | |
| inform | 4.77(Sd=1.09) | **4.91**(Sd=1.23) | **5.47***(Sd=0.81) | 5.27(Sd=1.02) | **5.29***(Sd=0.94) | 5.16(Sd=1.07) |
| natural | **4.76**(Sd=1.26) | 4.67(Sd=1.25) | **4.99***(Sd=1.13) | 4.62(Sd=1.28) | **4.86**(Sd=1.13) | 4.74(Sd=1.23) |
| quality | **4.77**(Sd=1.19) | 4.54(Sd=1.28) | **4.54**(Sd=1.18) | 4.53(Sd=1.26) | 4.51(Sd=1.14) | **4.58**(Sd=1.33) |

Table 2: System performance per dataset ('*' denotes p<0.05).

the same pattern as WBMs, while the average similarity score (sim) seems to be related to human *quality* ratings.

Looking at GBMs, we observe that they seem to be related to *naturalness* and *quality* ratings. Less complex utterances, as measured by readability (RE) and word length (cpw), have higher *naturalness* ratings. More complex utterances, as measured in terms of their length (len), number of words (wps), syllables (sps, spw) and polysyllables (pol, ppw), have lower *quality* evaluation. Utterances measured as more grammatical are on average evaluated higher in terms of *naturalness*.

While these initial results may suggest a relation between automatic metrics and human ratings, average scores can be misleading, as they only provide a system-level overview but do not measure the strength of association on sentence-level. This leads us to inspect the correlation of human and automatic metrics for each MR-system output pair.

## 7 Relation of Human and Automatic Metrics

### 7.1 Human Correlation Analysis

We calculate the correlation between automatic metrics and human ratings using the Spearman coefficient ($\rho$). We split the data per dataset and system in order to make valid pairwise comparisons. To handle outliers within human ratings, we use the median score of the three human raters. Following Kilickaya et al. (2017), we use the Williams' test (Williams, 1959) to determine significant differences between correlations. Table 3 summarises the correlation results between automatic metrics and human ratings, listing the best (i.e., highest absolute $\rho$) results for each type of metric (see Appendix A, Table 9 for details). Our results suggest that:

• In sum, no metric produces an even moderate correlation with human ratings, independently of dataset, system, or aspect of human rating. This

contrasts with our initially promising results on the system-level (see Section 6) and will be further discussed in Section 8. Note that similar inconsistencies between document- and sentence-level evaluation results are observed in MT (Specia et al., 2010).

• Similar to our previous results in Section 6, we find that WBMs show better correlations to human ratings of *informativeness* whereas GBMs show better correlations to *quality* and *naturalness*.

• Human ratings for *informativeness*, *naturalness* and *quality* are highly correlated with each other, with the highest correlation between the latter two ($\rho = 0.81$) reflecting that they both target surface realisation.

• All WBMs produce similar results (see Figure 1 and 2): They are strongly correlated with each other, and most of them produce correlations with human ratings which are *not* significantly different from each other. GBMs, on the other hand, show greater diversity.

• Correlation results are system- and dataset-specific (also see Appendix A, Tables 10–11). We observe the highest correlation for TGEN on BAGEL (Figures 1 and 2) and LOLS on SFREST, whereas RNNLG often shows low correlation between metrics and human ratings. This lets us conclude that WBMs and GBMs are sensitive to different systems and datasets.

• The highest positive correlation is observed between the number of words (wps) and *informativeness* for the TGEN system on BAGEL ($\rho = 0.33$, $p < 0.01$, see Figure 1). However, the wps metric (amongst most others) is not robust across systems and datasets: Its correlation on other datasets is very weak, ($\rho \leq .18$) and its correlation with informativeness ratings of LOLS's output is insignificant.

• As a sanity check, we also measure a random score $[0.0, 1.0]$ which proves to have a close-to-zero correlation with human ratings (highest $\rho = 0.09$).

| | | BAGEL | | SFHOTEL | | SFREST | |
|---|---|---|---|---|---|---|---|
| | | TGEN | LOLS | RNNLG | LOLS | RNNLG | LOLS |
| Best WBM | inform | 0.30* (B1) | 0.20* (RG) | 0.09 (B1) | 0.14* (LEP) | 0.13* (SIM) | 0.28* (LEP) |
| | natural | -0.19* (TER) | -0.19* (TER) | 0.10* (MET) | -0.20* (TER) | 0.17* (RG) | 0.19* (MET) |
| | quality | -0.16* (TER) | 0.16* (MET) | 0.10* (MET) | -0.12* (TER) | 0.09* (MET) | 0.18* (LEP) |
| Best GBM | inform | 0.33* (WPS) | 0.16* (PPW) | -0.09 (PPW) | 0.13* (CPW) | 0.11* (LEN) | 0.21* (LEN) |
| | natural | -0.25* (LEN) | -0.28* (WPS) | -0.17* (LEN) | -0.18* (SPS) | -0.19* (WPS) | -0.21* (SPS) |
| | quality | -0.19* (CPW) | -0.31* (PRS) | -0.16* (PPW) | -0.17* (SPW) | -0.11* (PRS) | -0.16* (SPS) |

Table 3: Spearman correlation between metrics and human ratings, with '*' denoting $p < 0.05$. Best metrics show the highest absolute value of $\rho$.

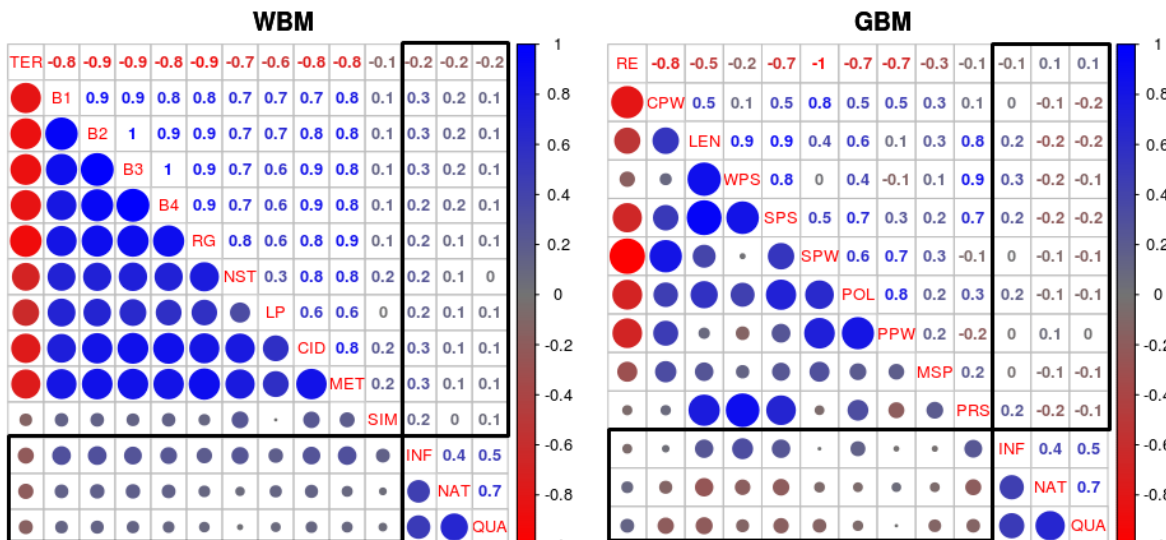

Figure 1: Spearman correlation results for TGEN on BAGEL. Bordered area shows correlations between human ratings and automatic metrics, the rest shows correlations among the metrics.

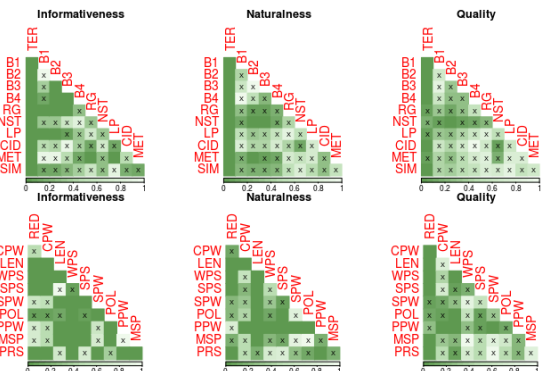

Figure 2: Williams test results: X represents a *non*-significant difference between correlations ($p < 0.05$; top: WBMs, bottom: GBMs).

## 7.2 Accuracy of Relative Rankings

We now evaluate a more coarse measure, namely the metrics' ability to predict relative human ratings. That is, we compute the score of each metric for two system output sentences corresponding to the same MR. The prediction of a metric is correct if it orders the sentences in the same way as median human ratings (note that ties are allowed). Following from previous work (Vedantam et al.,

2015; Kilickaya et al., 2017), we mainly concentrate on WBMs. Results summarised in Table 4 show that most metrics' performance is not significantly different from that of a random score (Wilcoxon signed rank test), see details in Appendix A, Table 12. While the random score fluctuates between 25.4–44.5% prediction accuracy, the metrics achieve an accuracy of between 30.6–49.8%. Again, the performance of the metrics is dataset-specific: Metrics perform best on BAGEL data; for SFHOTEL, metrics show mixed performance; while for SFREST, metrics perform worst.

**Discussion:** Our data differs from the one used in previous work (Vedantam et al., 2015; Kilickaya et al., 2017), which uses explicit relative rankings ("*Which output do you prefer?*"), whereas we compare two Likert-scale ratings. As such, we have 3 possible outcomes (allowing ties). This way, we can account for equally valid system outputs, which is one of the main drawbacks of forced-choice approaches (Hodosh and Hockenmaier, 2016). In relation to previous work, our results are reasonable: Kilickaya et al. (2017) report results between 60-74% accuracy for binary

|  |  | informat. | naturalness | quality |
|---|---|---|---|---|
| BAGEL | raw data | TER, BLEU1-4, ROUGE, NIST, LEPOR, CIDEr, METEOR, SIM | TER, BLEU1-4, ROUGE, NIST, LEPOR, CIDEr, METEOR, SIM | TER, BLEU1-4, ROUGE, NIST, LEPOR, CIDEr, METEOR, SIM |
| SFHOTEL | raw data | TER, BLEU1-4, ROUGE, LEPOR, CIDEr, METEOR, SIM | METEOR | N/A |
| SFREST | raw data | SIM | LEPOR | N/A |
|  | quant. data | TER, BLEU1-4, ROUGE, NIST, LEPOR, CIDEr, METEOR SIM | N/A | N/A |

Table 4: Metrics predicting relative human rating with significantly higher than random accuracy.

classification on machine-machine data, which is comparable to our results for 3-way classification.

Also, we observe a mismatch between the ordinal human ratings and the continuous metrics. For example, humans might rate system A and system B both as a 6, whereas BLEU, for example, might assign 0.98 and 1.0 respectively, meaning that BLEU will declare system B as the winner. In order to account for this mismatch, we quantise our metric data to the same scale as the median scores from our human ratings.[7] Applied to SFREST, where we previously got our worst results, we can see an improvement for predicting *informativeness*, where all metrics now perform significantly better than the random baseline, see Table 4. In future, we will investigate related discriminative approaches, e.g. (Hodosh and Hockenmaier, 2016; Kannan and Vinyals, 2017), where the task is simplified to distinguishing correct from incorrect output.

## 8 Error Analysis

In this section, we attempt to uncover why automatic metrics perform so poorly.

### 8.1 Scales

We first explore the hypothesis that metrics are good in distinguishing extreme cases, i.e. system outputs which are rated as clearly good or bad by the human judges, but do not perform well for utterances rated in the middle of the Likert scale, as suggested by Kilickaya et al. (2017). We 'bin' our data into three groups: *bad*, which comprises low

---

[7]Note that this mismatch can also be accounted for by continuous rating scales, as suggested by Belz and Kow (2011).

ratings ($\leq 2$); *good*, comprising high ratings ($\geq 5$); and finally a group comprising *average* ratings. We find that utterances with low human ratings of *informativeness* and *naturalness* correlate significantly better ($p < 0.05$) with automatic metrics than those with average and good human ratings. For example, as shown in Figure 3, the correlation between WBMs and human ratings for utterances with low *informativeness* scores ranges between $0.3 \leq \rho \leq 0.5$ (moderate correlation), while the highest correlation for utterances of average and high informativeness barely reaches $\rho \leq 0.2$ (very weak correlation). The same pattern can be observed for correlations with *quality* and *naturalness* ratings (see Appendix A, Table 13).

This discrepancy in correlation results between low and other user ratings, together with the fact that the majority of system outputs are rated "good" for informativeness (79%), naturalness (64%) and quality (58%), whereas low ratings do not exceed 7% in total, could explain why the overall correlations are low (Section 7) despite the observed trends in relationship between average system-level performance scores (Section 6). It also explains why the RNNLG system, which contains very few instances of low user ratings, shows poor correlation between human ratings and automatic metrics.

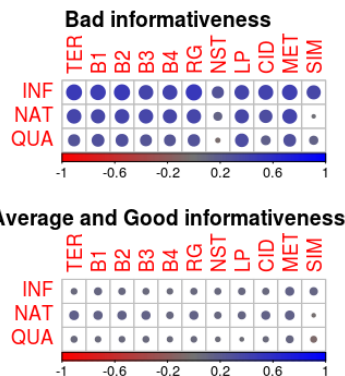

Figure 3: Correlation between automatic metrics (WBMs) and human ratings for utterances of bad informativeness (top), and average and good informativeness (bottom).

### 8.2 Impact of Target Data

**Type of Data:** In Section 7.1, we observed that datasets have a significant impact on how well automatic metrics reflect human ratings. A closer inspection shows that BAGEL data differs significantly from SFREST and SFHOTEL, both in terms

of grammatical and MR properties. BAGEL has significantly shorter references both in terms of number of characters and words compared to the other two datasets. Although being shorter, the words in BAGEL references are significantly more often polysyllabic. Furthermore, BAGEL only consists of utterances generated from *inform* MRs, while SFREST and SFHOTEL also have less complex MR types, such as *confirm*, *goodbye*, etc. Utterances produced from *inform* MRs are significantly longer and have a significantly higher correlation with human ratings of *informativeness* and *naturalness*, than *non-inform* type of utterances (see Appendix A, Table 14). In other words, BAGEL is the most complex dataset to generate from, but also the one where metrics perform most reliably (note that the correlation is still only weak). One possible explanation of this discrepancy is that BAGEL only contains 2 human references per MR, whereas SFHOTEL and SFREST both contain 5.35 references per MR on average. Having more references means that WBMs naturally will return higher scores ('anything goes'). This problem could possibly be solved by weighting multiple references according to their quality, as suggested by (Galley et al., 2015), or following a reference-less approach (Specia et al., 2010).

**Quality of Data:** Our corpora contain crowd-sourced human references that have grammatical errors, e.g. "*Fifth Floor does not allow childs*" (SFREST reference), or tautology issues, e.g. "*Do you want a hotel in the pricey price range?*" (SFHOTEL reference). Corpus-based methods may pick up these errors, and word-based metrics will rate these system utterances as correct, whereas we can expect human judges to be sensitive to ungrammatical utterances. Note that the parsing score (while being a crude approximation of grammaticality) achieves one of our highest correlation results against human ratings, with $|\rho| = .31$. Grammatical errors raise questions about the quality of the training data, especially when being crowd-sourced. For example, Belz and Reiter (2006) find that human experts assign low rankings to their original corpus text. Again, weighting (Galley et al., 2015) or reference-less approaches (Specia et al., 2010) might remedy this issue.

## 9 Combined Model

In this section, we present our new metric, which we call RAINBOW to reflect that it combines mul-

tiple other metrics. Recent results in MT (Yu et al., 2015; Bojar et al., 2016) show that combining multiple metrics can achieve the best overall correlation with human ratings at the sentence-level. This motivates us to combine the respective strengths of WBMs and GBMs into a single model using ensemble learning (Random Forest, RF) (Breiman, 2001).

**Setup:** We use quantised metrics scores, as described in Section 7.2, and median human ratings, with a 70/30% split for training and testing and 10-fold cross-validation on the training data to tune the optimal number of predictors selected for growing trees. 100 trees were grown with 2 variables randomly sampled as candidates at each split. We investigate four different models: 1) all WBMs and GBMs metrics combined as predictors (RAINBOW **WBM+GBM**), 2) all WBMs combined (RAINBOW **WBM**), 3) all GBMs combined (RAINBOW **GBM**), and 4) only top-5 predictors combined (RAINBOW **Top5**).

**Results:** The results in Table 5 show that the combined metrics are both able to increase the correlation with human ratings and behave in a robust way across different datasets and systems. The highest correlation is achieved when all the automatic metrics are combined ($0.71 \leq \rho \leq 0.81$). The models which only combine WBMs or only combine GBMs correlate significantly worse with human ratings ($p < 0.001$, Williams test). This supports our hypothesis that a combination of WBMs and GBMs can overcome the weaknesses of both.

Computing a full set of automatic metrics (21 in our case) as the input for the ensemble model is time consuming, which is especially problematic if the metric is used as a loss function, as e.g. in (Lampouras and Vlachos, 2016; Li et al., 2017). To overcome this limitation, we select the top 5 features (according to the RF model), using a recursive feature elimination. The top 5 features for predicting *informativeness* are len, SIM, METEOR, cpw and ROUGE; for *naturalness* – wps, METEOR, cpw, sps and TER; and for *quality* – METEOR, ROUGE, sps, BLEU2 and BLEU4. Overall, the correlation of RAINBOW Top5 can be classified as moderate, except for RNNLG and TGEN, where it correlates weakly with naturalness (0.35–0.38). While this model performs significantly worse than RAINBOW WBM+GBM and RAINBOW WBM, it significantly outperforms all of the single metrics, and

|  |  | BAGEL | | SFHOTEL | | SFREST | |
| --- | --- | --- | --- | --- | --- | --- | --- |
|  |  | TGEN | LOLS | RNNLG | LOLS | RNNLG | LOLS |
| RAINBOW WBM+GBM | inform | 0.71* | 0.76* | 0.71* | 0.71* | 0.71* | 0.77* |
|  | natural | 0.74* | 0.75* | 0.79* | 0.79* | 0.73* | 0.79* |
|  | quality | 0.71* | 0.79* | 0.79* | 0.75* | 0.72* | 0.81* |
| RAINBOW WBM | inform | 0.67* | 0.75* | 0.61* | 0.67* | 0.65* | 0.72* |
|  | natural | 0.62* | 0.73* | 0.69* | 0.77* | 0.64* | 0.72* |
|  | quality | 0.67* | 0.74* | 0.66* | 0.75* | 0.62* | 0.77* |
| RAINBOW GBM | inform | 0.57* | 0.46* | 0.51* | 0.56* | 0.52* | 0.58* |
|  | natural | 0.49* | 0.55* | 0.57* | 0.60* | 0.54* | 0.62* |
|  | quality | 0.50* | 0.59* | 0.54* | 0.52* | 0.54* | 0.65* |
| RAINBOW Top5 | inform | 0.59* | 0.50* | 0.47* | 0.60* | 0.54* | 0.67* |
|  | natural | 0.35* | 0.50* | 0.36* | 0.58* | 0.38* | 0.52* |
|  | quality | 0.41* | 0.42* | 0.40* | 0.58* | 0.39* | 0.59* |

Table 5: Spearman correlation between metrics and human ratings for combined models ('*' = $p < 0.05$).

| Study/ task | Sentence Planning | Surface Realisation | Domain |
| --- | --- | --- | --- |
| this paper | weak positive ($\rho = 0.33$, WPS) | weak negative ($\rho = 0. - 31$, parser) | NLG, restaurant/hotel search |
| (Reiter and Belz, 2009) | none | strong positive (Pearson's $r = 0.96$, NIST) | NLG, weather forecast |
| (Stent et al., 2005) | weak positive ($\rho = 0.47$, LSA) | negative ($\rho = -0.56$, NIST) | paraphrasing of news |
| (Liu et al., 2016) | weak positive ($\rho = 0.35$, BLEU-4) | N/A | dialogue/Twitter pairs |
| (Elliott and Keller, 2014) | positive ($\rho = 0.53$, METEOR) | N/A | image caption |
| (Kilickaya et al., 2017) | positive ($\rho = 0.64$, SPICE) | N/A | image caption |

Table 6: Best correlation results achieved by our and previous work.

it shows comparable performance to the RAINBOW GBM model, which combines more than twice as many features. However, in contrast to the RAINBOW GBM model, the Top5 model still needs costly human reference texts (each Top5 model contains at least one WBM). Thus, a promising future direction is reference-less quality prediction, as used in MT, e.g. (Specia et al., 2010).

## 10 Related Work

Table 6 summarises results published by previous studies in related fields investigating the relation between human scores and automatic metrics. These studies mainly considered WBMs, while we are the first study to consider GBMs. Some studies ask users to provide separate ratings for surface realisation (e.g. asking about 'clarity' or 'fluency'), whereas other studies focus only on sentence planning (e.g. 'accuracy', 'adequacy', or 'correctness'). In general, correlations reported by previous work range from weak to strong. The results confirm that correlations and metric performance appear to be system- and dataset-specific, with some results even directly opposed to each other, e.g. (Reiter and Belz, 2009) and (Stent et al., 2005). There is a general trend showing that best-performing metrics tend to be the more complex ones, combining word-overlap, semantic similarity and term frequency weighting. (Note, however, that most previous works do not report whether any of the metric correlations are significantly dif-

ferent from each other.)

## 11 Conclusions and Future Directions

This paper shows that state-of-the-art automatic evaluation metrics for NLG systems do not sufficiently reflect human ratings. Word-based metrics make two strong assumptions: They treat human-generated references as a gold-standard, which is *correct* and *complete*. We argue that these assumptions are invalid for corpus-based NLG, especially when using crowd-sourced datasets. Grammar-based metrics, on the other hand, do not rely on human-generated references and are not influenced by their quality. However, these metrics can be easily manipulated with grammatically-correct and easily-readable output that is unrelated to the input. To merge the advantages of WBMs and GBMs, we present a combined model, RAINBOW, which significantly improves correlation with human ratings.

In our future work, we will investigate more advanced metrics, as used in related fields, including: assessing output quality within the dialogue context, e.g. (Dušek and Jurčíček, 2016); extrinsic evaluation metrics, such as NLG's contribution to task success, e.g. (Rieser et al., 2014; Gkatzia et al., 2016; Hastie et al., 2016); and reference-less quality prediction as used in MT, e.g. (Specia et al., 2010).

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

## Appendix A: Detailed Results

| BAGEL | | | SFHOTEL | | | SFREST | | |
|---|---|---|---|---|---|---|---|---|
| Inf: 0.16* | Nat: 0.36* | Qua: 0.38* | Inf: 0.41* | Nat: 0.47* | Qua: 0.52* | Inf: 0.35* | Nat: 0.29* | Qua: 0.35* |
| TGEN: 0.42* | LOLS: 0.24* | | RNNLG: 0.52* | LOLS: 0.45* | | RNNLG: 0.42* | LOLS: 0.24* | |
| Total BAGEL: 0.31* | | | Total SFHOTEL: 0.50* | | | Total SFREST: 0.31* | | |
| Total all data: 0.45* | | | | | | | | |

Table 7: Intra-class correlation coefficient (ICC) for human ratings across the three datasets. "*" denotes statistical significance ($p < 0.05$).

| | BAGEL | | SFHOTEL | | SFREST | |
|---|---|---|---|---|---|---|
| | TGEN | LOLS | RNNLG | LOLS | RNNLG | LOLS |
| *metric* | *Avg / StDev* | *Avg / StDev* | *Avg / StDev* | *Avg / StDev* | *Avg / StDev* | *Avg / StDev* |
| TER | 0.36/0.24 | 0.33/0.24 | 0.28*/0.27 | 0.65*/0.32 | 0.41*/0.35 | 0.65*/0.27 |
| BLEU1 | 0.75*/0.21 | 0.81*/0.16 | 0.85*/0.18 | 0.66*/0.23 | 0.73*/0.24 | 0.59*/0.23 |
| BLEU2 | 0.68/0.23 | 0.72/0.21 | 0.78*/0.25 | 0.54*/0.28 | 0.62*/0.31 | 0.45*/0.29 |
| BLEU3 | 0.60/0.28 | 0.63/0.26 | 0.69*/0.32 | 0.42*/0.33 | 0.52*/0.37 | 0.34*/0.33 |
| BLEU4 | 0.52/0.32 | 0.53/0.33 | 0.56*/0.40 | 0.28*/0.33 | 0.44*/0.41 | 0.24*/0.32 |
| ROUGE | 0.76/0.18 | 0.78/0.17 | 0.83*/0.18 | 0.64*/0.21 | 0.72*/0.24 | 0.58*/0.22 |
| NIST | 4.44*/2.05 | 4.91*/2.04 | 4.37*/2.19 | 3.49*/1.99 | 4.86/2.55 | 4.01*/2.07 |
| LEPOR | 0.46/0.22 | 0.50*/0.19 | 0.52*/0.23 | 0.30*/0.16 | 0.51*/0.25 | 0.30*/0.17 |
| CIDEr | 2.92/2.40 | 3.01/2.27 | 3.08*/2.05 | 1.66*/1.67 | 3.39*/2.53 | 2.09*/1.73 |
| METEOR | 0.50/0.22 | 0.53/0.23 | 0.62*/0.27 | 0.44*/0.20 | 0.54*/0.28 | 0.41*/0.19 |
| SIM | 0.66/0.09 | 0.65/0.12 | 0.76*/0.15 | 0.73*/0.14 | 0.76/0.13 | 0.77/0.14 |
| RE | 86.79/19.48 | 83.39/20.41 | 70.90/17.07 | 69.62/19.14 | 64.67/19.07 | 64.27/22.22 |
| msp | 0.04*/0.21 | 0.14*/0.37 | 0.68/0.78 | 0.69/0.77 | 0.78/0.82 | 0.85/0.89 |
| prs | 84.51*/25.78 | 93.30*/27.04 | 97.58*/32.58 | 107.90*/36.41 | 93.74/34.98 | 97.20/39.30 |
| len | 38.20*/14.22 | 42.54*/14.11 | 49.06*/15.77 | 51.69*/17.30 | 53.27*/19.50 | 50.92*/18.74 |
| wps | 10.08*/3.10 | 10.94*/3.19 | 11.43*/3.63 | 12.07*/4.17 | 11.15*/4.37 | 10.52*/4.21 |
| sps | 13.15*/4.98 | 14.61*/5.13 | 16.03*/4.88 | 17.02*/5.90 | 16.39*/6.17 | 15.41*/5.92 |
| cpw | 3.77/0.60 | 3.88/0.59 | 4.34/0.58 | 4.36/0.63 | 4.86*/0.64 | 4.94*/0.76 |
| spw | 1.30/0.22 | 1.33/0.23 | 1.43/0.23 | 1.43/0.26 | 1.50/0.26 | 1.50/0.29 |
| pol | 2.22/1.21 | 2.40/1.16 | 1.24/1.04 | 1.33/1.04 | 1.69/1.12 | 1.57/1.07 |
| ppw | 0.22/0.09 | 0.22/0.09 | 0.11/0.10 | 0.12/0.09 | 0.16/0.11 | 0.16/0.12 |
| informativeness | 4.77/1.09 | 4.91/1.23 | 5.47*/0.81 | 5.27/1.02 | 5.29*/0.94 | 5.16/1.07 |
| naturalness | 4.76/1.26 | 4.67/1.25 | 4.99*/1.13 | 4.62/1.28 | 4.86/1.13 | 4.74/1.23 |
| quality | 4.77/1.19 | 4.54/1.28 | 4.54/1.18 | 4.53/1.26 | 4.51/1.14 | 4.58/1.33 |

Table 8: The systems' performance for all datasets. *Avg* denotes a mean value, *StDev* stands for standard deviation, "*" denotes a statistically significant difference ($p < 0.05$) between the two systems on the given dataset.

| metric | BAGEL | | | | | | SFHOTEL | | | | | | SFREST | | | | | |
|---|---|---|---|---|---|---|---|---|---|---|---|---|---|---|---|---|---|---|
| | TGEN | | | LOLS | | | RNNLG | | | LOLS | | | RNNLG | | | LOLS | | |
| | inf | nat | qual | inf | nat | qual | inf | nat | qual | inf | nat | qual | inf | nat | qual | inf | nat | qual |
| TER | -0.21* | -0.19* | -0.16* | -0.16* | -0.19* | -0.16* | -0.03 | -0.09 | -0.08 | -0.06 | **-0.20*** | -0.12* | 0.02 | -0.14* | -0.08 | **-0.16*** | -0.14* | -0.14* |
| BLEU1 | **0.30*** | 0.15* | 0.13 | 0.13 | 0.15* | 0.13 | 0.09 | 0.09* | 0.08 | 0.01 | 0.12* | 0.06 | 0.02 | 0.12* | 0.06 | **0.19*** | 0.15* | 0.13* |
| BLEU2 | **0.30*** | 0.17* | 0.14 | 0.12 | 0.14* | 0.11 | 0.08 | 0.09* | 0.07 | 0.00 | 0.12* | 0.07 | 0.01 | 0.13* | 0.07 | **0.14*** | 0.10* | 0.08* |
| BLEU3 | **0.27*** | 0.17* | 0.12 | 0.11 | 0.13 | 0.10 | 0.06 | 0.08 | 0.06 | 0.01 | 0.11* | 0.08 | 0.02 | 0.13* | 0.09* | 0.12* | 0.08 | 0.07 |
| BLEU4 | **0.23*** | 0.15* | 0.11 | 0.11 | 0.13 | 0.10 | 0.06 | 0.05 | 0.07 | 0.00 | 0.02 | 0.03 | 0.03 | 0.12* | 0.07 | 0.12* | 0.04 | 0.05 |
| ROUGE | 0.20* | 0.11 | 0.09 | 0.20* | 0.17* | 0.15* | 0.07 | 0.09 | 0.08 | -0.01 | 0.04 | 0.02 | 0.04 | 0.17* | 0.09* | 0.12* | 0.11* | 0.08 |
| NIST | 0.24* | 0.07 | 0.02 | 0.16* | 0.13 | 0.11 | 0.07 | 0.05 | 0.01 | 0.02 | 0.14* | 0.11* | 0.03 | 0.07 | 0.01 | **0.15*** | 0.08 | 0.07 |
| LEPOR | **0.17*** | 0.12 | **0.07** | -0.07 | 0.02 | -0.04 | 0.03 | 0.03 | 0.03 | 0.14* | **0.17*** | 0.10* | 0.00 | 0.05 | -0.02 | **0.28*** | 0.17* | 0.18* |
| CIDEr | **0.26*** | 0.14* | 0.10 | 0.14* | 0.19* | 0.14* | 0.07 | 0.07 | 0.00 | 0.03 | 0.13* | 0.09 | 0.02 | 0.12* | 0.03 | 0.10* | 0.11* | 0.08 |
| METEOR | 0.29 | 0.09 | 0.09 | 0.20* | 0.18* | 0.16* | 0.07 | 0.10* | 0.10* | 0.05 | 0.06 | 0.04 | 0.06 | 0.16* | 0.09* | **0.23*** | 0.19* | 0.17* |
| SIM | 0.16* | 0.04 | 0.06 | 0.14* | 0.13 | 0.09 | -0.05 | -0.12* | -0.11* | 0.03 | -0.03 | -0.08 | 0.13* | -0.06 | -0.08* | 0.19* | 0.01 | 0.02 |
| RE | -0.06 | **0.09** | 0.13 | -0.09 | -0.04 | 0.04 | 0.00 | 0.03 | **0.10*** | -0.01 | 0.03 | 0.09 | 0.00 | -0.05 | 0.02 | 0.04 | **0.09*** | **0.08*** |
| cpw | 0.03 | -0.12 | **-0.19*** | 0.08 | 0.05 | -0.03 | 0.02 | -0.02 | **-0.09*** | **0.13*** | **0.14*** | 0.06 | 0.02 | 0.11* | 0.01 | 0.06 | 0.10* | 0.09* |
| len | 0.25* | -0.25* | -0.21* | 0.04 | -0.19* | -0.24* | 0.01 | -0.17* | -0.09 | 0.12* | -0.08 | -0.07 | 0.11* | -0.17* | -0.08 | 0.21* | -0.14* | -0.09* |
| wps | 0.33* | -0.17* | -0.12 | -0.05 | **-0.28*** | **-0.29*** | 0.01 | -0.15* | -0.05 | 0.08 | -0.12* | -0.08 | 0.11* | -0.19* | -0.07 | 0.18* | -0.15* | -0.11* |
| sps | 0.25* | -0.20* | -0.17* | 0.03 | -0.17* | -0.23* | -0.02 | -0.16* | -0.08 | 0.02 | -0.18* | -0.16* | 0.07 | -0.17* | -0.08 | 0.12* | -0.21* | -0.16* |
| spw | 0.01 | -0.07 | **-0.13** | 0.10 | **0.09** | 0.02 | -0.08 | -0.02 | -0.11* | -0.10* | -0.10* | -0.17* | -0.07 | 0.06 | -0.03 | -0.14* | **-0.10*** | -0.11* |
| pol | 0.16* | -0.06 | -0.07 | 0.11 | -0.03 | -0.12 | -0.07 | -0.10* | -0.15* | 0.01 | -0.09 | -0.14* | -0.04 | -0.04 | -0.03 | -0.02 | -0.13* | -0.11* |
| ppw | -0.02 | 0.06 | 0.00 | **0.16*** | 0.15* | 0.08 | -0.09 | -0.06 | -0.16* | -0.02 | -0.01 | -0.09 | -0.09* | **0.08** | 0.00 | -0.13* | -0.05 | -0.07 |
| msp | -0.02 | -0.06 | -0.11 | 0.02 | -0.02 | -0.10 | -0.01 | -0.10* | -0.08 | 0.05 | -0.02 | -0.03 | 0.05 | 0.02 | -0.06 | 0.12* | 0.01 | 0.07 |
| prs | 0.23* | -0.18* | -0.13 | -0.05 | -0.24* | -0.31* | 0.02 | -0.13* | -0.09 | 0.13* | -0.05 | -0.04 | 0.11* | -0.15* | -0.11* | 0.16* | -0.20* | -0.16* |

Table 9: Spearman correlation between metrics and human ratings for individual datasets and systems. "*" denotes statistically significant correlation ($p < 0.05$), bold font denotes significantly stronger correlation when comparing two systems on the same dataset.

| | BAGEL | | | SFHOTEL | | | SFREST | | |
|---|---|---|---|---|---|---|---|---|---|
| | *inf* | *nat* | *qual* | *inf* | *nat* | *qual* | *inf* | *nat* | *qual* |
| TER | -0.19* | -0.19* | -0.15* | -0.10* | -0.19* | -0.07* | -0.09* | -0.15* | -0.08* |
| BLEU1 | 0.23* | 0.14* | 0.11* | 0.11* | 0.18* | 0.07* | 0.11* | 0.14* | 0.07* |
| BLEU2 | 0.21* | 0.15* | 0.12* | 0.10* | 0.17* | 0.07* | 0.09* | 0.13* | 0.06* |
| BLEU3 | 0.19* | 0.15* | 0.11* | 0.09* | 0.16* | 0.07* | 0.08* | 0.12* | 0.06* |
| BLEU4 | 0.18* | 0.14* | 0.10* | 0.08* | 0.10* | 0.06 | 0.09* | 0.09* | 0.05 |
| ROUGE | 0.20* | 0.13* | 0.11* | 0.09* | 0.15* | 0.06 | 0.09* | 0.15* | 0.06* |
| NIST | 0.21* | 0.09 | 0.06 | 0.07* | 0.13* | 0.06 | 0.10* | 0.08* | 0.03 |
| LEPOR | 0.07 | 0.07 | 0.01 | 0.13* | 0.15* | 0.05 | 0.16* | 0.12* | 0.04 |
| CIDEr | 0.21* | 0.16* | 0.12* | 0.10* | 0.16* | 0.05 | 0.08* | 0.12* | 0.04 |
| METEOR | 0.25* | 0.13* | 0.12* | 0.11* | 0.15* | 0.08* | 0.15* | 0.18* | 0.11* |
| SIM | 0.15* | 0.09 | 0.07 | 0.01 | -0.04 | -0.09* | 0.15* | -0.02 | -0.02 |
| RE | -0.08 | 0.03 | 0.09 | 0.01 | 0.04 | 0.10* | 0.02 | 0.02 | 0.06 |
| cpw | 0.05 | -0.04 | -0.12* | 0.07* | 0.05 | -0.02 | 0.04 | 0.10* | 0.06 |
| len | 0.14* | -0.22* | -0.24* | 0.05 | -0.14* | -0.07* | 0.16* | -0.15* | -0.09* |
| wps | 0.14* | -0.23* | -0.23* | 0.03 | -0.14* | -0.06 | 0.14* | -0.17* | -0.10* |
| sps | 0.14* | -0.19* | -0.21* | -0.01 | -0.18* | -0.12* | 0.10* | -0.18* | -0.12* |
| spw | 0.05 | 0.00 | -0.06 | -0.10* | -0.06 | -0.14* | -0.11* | -0.02 | -0.07* |
| pol | 0.13* | -0.05 | -0.10* | -0.04 | -0.10* | -0.14* | -0.03 | -0.08* | -0.08* |
| ppw | 0.06 | 0.11* | 0.04 | -0.06 | -0.04 | -0.13* | -0.11* | 0.01 | -0.04 |
| msp | 0.02 | -0.04 | -0.11* | 0.02 | -0.06 | -0.06 | 0.08* | 0.01 | 0.01 |
| prs | 0.10 | -0.22* | -0.25* | 0.05 | -0.12* | -0.07 | 0.13* | -0.18* | -0.13* |

Table 10: Spearman correlation between metrics and human ratings for each dataset. "*" denotes statistically significant correlation ($p < 0.05$).

| | TGEN | | | LOLS | | | RNNLG | | |
|---|---|---|---|---|---|---|---|---|---|
| | *inf* | *nat* | *qual* | *inf* | *nat* | *qual* | *inf* | *nat* | *qual* |
| TER | -0.21* | -0.19* | -0.16* | -0.07* | -0.15* | -0.11* | -0.02 | -0.13* | -0.08* |
| BLEU1 | 0.30* | 0.15* | 0.13 | 0.08* | 0.12* | 0.08* | 0.07* | 0.13* | 0.07* |
| BLEU2 | 0.30* | 0.17* | 0.14 | 0.05 | 0.11* | 0.07* | 0.06* | 0.14* | 0.08* |
| BLEU3 | 0.27* | 0.17* | 0.12 | 0.04 | 0.09* | 0.07* | 0.06 | 0.13* | 0.08* |
| BLEU4 | 0.23* | 0.15* | 0.11 | 0.04 | 0.04 | 0.04 | 0.06 | 0.11* | 0.08* |
| ROUGE | 0.20* | 0.11 | 0.09 | 0.05 | 0.09* | 0.05 | 0.07* | 0.15* | 0.09* |
| NIST | 0.25* | 0.07 | 0.02 | 0.07* | 0.11* | 0.09* | 0.04 | 0.06* | 0.01 |
| LEPOR | 0.17* | 0.12 | 0.07 | 0.13* | 0.13* | 0.11* | 0.02 | 0.05 | 0.00 |
| CIDEr | 0.26* | 0.14* | 0.10 | 0.05 | 0.13* | 0.09* | 0.04 | 0.10* | 0.02 |
| METEOR | 0.29* | 0.09 | 0.09 | 0.14* | 0.13* | 0.12* | 0.08* | 0.15* | 0.10* |
| SIM | 0.16* | 0.04 | 0.06 | 0.14* | 0.02 | 0.00 | 0.05 | -0.08* | -0.09* |
| RE | -0.06 | 0.09 | 0.13 | -0.02 | 0.04 | 0.07* | 0.02 | -0.01 | 0.06* |
| cpw | 0.03 | -0.12 | -0.19* | 0.11* | 0.11* | 0.08* | -0.02 | 0.02 | -0.05 |
| len | 0.25* | -0.25* | -0.21* | 0.17* | -0.12* | -0.10* | 0.06 | -0.18* | -0.08* |
| wps | 0.33* | -0.17* | -0.12 | 0.11* | -0.17* | -0.13* | 0.07* | -0.17* | -0.06 |
| sps | 0.25* | -0.20* | -0.17* | 0.09* | -0.19* | -0.17* | 0.03 | -0.17* | -0.08* |
| spw | 0.01 | -0.07 | -0.13 | -0.07* | -0.06 | -0.10* | -0.09* | 0.01 | -0.07* |
| pol | 0.16* | -0.06 | -0.07 | -0.02 | -0.09* | -0.11* | -0.08* | -0.08* | -0.09* |
| ppw | -0.02 | 0.06 | 0.00 | -0.08* | 0.00 | -0.05 | -0.11* | 0.00 | -0.07* |
| msp | -0.02 | -0.06 | -0.11 | 0.10* | 0.00 | 0.02 | 0.02 | -0.04 | -0.07* |
| prs | 0.23* | -0.18* | -0.13 | 0.12* | -0.16* | -0.15* | 0.07* | -0.14* | -0.10* |

Table 11: Spearman correlation between metrics and human ratings for each system. "*" denotes statistical significance ($p < 0.05$).

| Accuracy | rand | TER | BLEU1 | BLEU2 | BLEU3 | BLEU4 | ROUGE | NIST | LEPOR | CIDEr | METE | RE | SIM |
|---|---|---|---|---|---|---|---|---|---|---|---|---|---|
| **BAGEL** | | | | | | | | | | | | | |
| inform | 37.13 | 45.05* | 41.58* | 41.58* | 42.57* | 42.08* | 43.07* | 43.07* | 41.58* | 43.07* | 45.54* | 37.13 | 41.09* |
| natural | 42.08 | 47.03* | 46.04* | 45.54* | 44.06* | 45.05* | 46.04* | 44.55* | 46.53* | 45.05* | 45.05* | 42.08 | 43.07* |
| quality | 33.17 | 45.54* | 43.07* | 40.10* | 40.59* | 43.56* | 43.07* | 41.09* | 40.59* | 42.08* | 41.58* | 37.62 | 42.57* |
| **SFHotel** | | | | | | | | | | | | | |
| inform | 25.38 | 34.92* | 35.68* | 35.18* | 35.68* | 34.67* | 36.43* | 31.41 | 32.16* | 33.92* | 36.43* | 34.92* | 33.92* |
| natural | 41.96 | 45.73 | 46.48 | 45.48 | 46.48 | 45.23 | 48.74 | 41.21 | 43.72 | 44.72 | 49.75* | 37.19 | 46.98 |
| quality | 44.47 | 40.95 | 40.95 | 42.21 | 44.72 | 41.46 | 43.22 | 40.2 | 40.95 | 42.46 | 45.98 | 33.67 | 37.44 |
| **SFRest** | | | | | | | | | | | | | |
| inform | 33.68 | 36.27 | 35.41 | 34.02 | 34.72 | 36.96 | 33.16 | 35.58 | 36.27 | 32.47 | 34.72 | 38.34* | 42.66* |
| natural | 36.10 | 40.41 | 40.07 | 38.86 | 38.34 | 38.86 | 38.17 | 39.38 | 41.11* | 36.79 | 39.38 | 39.38 | 38.00 |
| quality | 39.38 | 37.13 | 36.96 | 39.21 | 37.65 | 39.55 | 36.10 | 38.69 | 39.72 | 35.23 | 34.89 | 40.93 | 37.31 |
| **SFRest, quant.** | | | | | | | | | | | | | |
| inform | 31.95 | 35.75* | 36.27* | 34.37* | 35.92* | 34.54* | 36.44* | 39.55* | 37.13* | 36.27* | 36.79* | 38.17* | 42.83* |
| quality | 39.21 | 33.33 | 34.37 | 32.3 | 30.57 | 26.94 | 34.54 | 33.16 | 35.92 | 30.92 | 31.61 | 32.47 | 35.41 |
| naturalness | 37.13 | 37.82 | 38.69 | 36.1 | 35.75 | 32.3 | 36.96 | 39.21 | 38.86 | 35.23 | 38.34 | 34.2 | 36.1 |

Table 12: Accuracy of metrics predicting relative human ratings, with "*" denoting statistical significance ($p < 0.05$).

| | informativeness | | naturalness | | quality | |
|---|---|---|---|---|---|---|
| | Bad | Good and avg | Bad | Good and avg | Bad | Good and avg |
| TER | **0.48*** | 0.07* | **0.31*** | 0.15* | 0.08 | 0.06* |
| BLEU1 | **0.45*** | 0.11* | **0.26*** | 0.13* | 0.07 | 0.04 |
| BLEU2 | **0.49*** | 0.09* | **0.29*** | 0.13* | 0.05 | 0.04* |
| BLEU3 | **0.40*** | 0.08* | **0.25*** | 0.13* | 0.01 | 0.05* |
| BLEU4 | **0.41*** | 0.07* | **0.21*** | 0.08* | 0.01 | 0.04 |
| ROUGE | **0.50*** | 0.08* | **0.28*** | 0.13* | 0.07 | 0.04* |
| NIST | **0.26** | 0.08* | **0.23*** | 0.08* | 0.08 | 0.03 |
| LEPOR | **0.40*** | 0.09* | **0.23*** | 0.10* | 0.03 | 0.01 |
| CIDEr | **0.42*** | 0.09* | **0.21*** | 0.12* | 0.02 | 0.04 |
| METEOR | **0.45*** | 0.14* | 0.24* | 0.15* | 0.03 | 0.08* |
| SIM | **0.37*** | 0.12* | **0.29*** | -0.03 | **0.21*** | -0.08* |

Table 13: Spearman correlation between WBM scores and human ratings for utterances from the *Bad* bin and utterances from the *Good* and *Average* bins. "*" denotes statistically significant correlation ($p < 0.05$), bold font denotes significantly stronger correlation for the *Bad* bin compared to the *Good* and *Average* bins.

| | informativeness | | naturalness | | quality | |
|---|---|---|---|---|---|---|
| | Inform | Not inform | Inform | Not inform | Inform | Not inform |
| TER | -0.08* | -0.10 | -0.17* | -0.18* | -0.09* | -0.11* |
| BLEU1 | 0.11* | 0.09 | 0.14* | 0.20* | 0.07* | 0.11* |
| BLEU2 | 0.09* | 0.10 | 0.14* | 0.20* | 0.07* | 0.13* |
| BLEU3 | 0.07* | 0.11* | 0.13* | 0.20* | 0.06* | 0.14* |
| BLEU4 | 0.06* | 0.11* | 0.09* | 0.18* | 0.05* | 0.14* |
| ROUGE | 0.08* | 0.12* | 0.14* | 0.22* | 0.06* | 0.16* |
| NIST | 0.08* | 0.05 | 0.10* | 0.06 | 0.07* | -0.06 |
| LEPOR | 0.09* | 0.16* | 0.11* | 0.16* | 0.05* | 0.04 |
| CIDEr | 0.10* | 0.01 | **0.16*** | 0.04 | **0.07*** | 0.02 |
| METEOR | 0.14* | 0.17* | 0.15* | 0.22* | 0.09* | 0.18* |
| SIM | 0.15* | 0.09 | -0.01 | -0.03 | -0.05* | -0.10 |
| cpw | **0.12*** | -0.15* | **0.09*** | -0.14* | **0.01** | -0.11* |
| len | 0.17* | 0.08 | -0.15* | -0.12* | -0.12* | -0.05 |
| wps | 0.11* | 0.19* | **-0.19*** | -0.03 | **-0.12*** | 0.01 |
| sps | 0.09* | 0.18* | **-0.20*** | -0.02 | **-0.17*** | 0.02 |
| spw | -0.06* | 0.09 | -0.03 | 0.01 | **-0.12*** | 0.01 |
| pol | **-0.08*** | 0.05 | -0.10* | -0.03 | -0.09* | -0.03 |
| ppw | **-0.14*** | -0.01 | 0.00 | -0.03 | -0.03 | -0.05 |
| msp | **0.11*** | -0.03 | 0.00 | -0.08 | -0.03 | -0.08 |
| prs | 0.10* | 0.18* | **-0.18*** | -0.04 | **-0.15*** | -0.02 |

Table 14: Spearman correlation between automatic metrics and human ratings for utterances of the *inform* MR type and utterances of other MR types. "*" denotes statistically significant correlation ($p < 0.05$), bold font denotes significantly stronger (absolute) correlation for *inform* MRs compared to non-*inform* MRs.

