# Peer review of "From BLEU to RAINBOW: Why We Need New Metrics for NLG."

_ACL 2017 — decision unknown_

[Official Review · Reviewer 1 · rating 3 · confidence 3]
soundness 5 · originality 5 · clarity 4 · impact 3 · substance 4 · appropriateness 5 · meaningful comparison 3 · presentation format Poster

- Strengths:

The paper addresses a long standing problem concerning automatic evaluation of
the output of generation/translation systems.

The analysis of all the available metrics is thorough and comprehensive.

The authors demonstrate a new metric with a higher correlation with human
judgements

The bibliography will help new entrants into the field.

- Weaknesses:

The paper is written as a numerical analysis paper, with very little insights
to linguistic issues in generation, the method of generation, the differences
in the output from a different systems and human generated reference.

It is unclear if the crowd source generated references serve well in the
context of an application that needs language generation.

- General Discussion:

Overall, the paper could use some linguistic examples (and a description of the
different systems) at the risk of dropping a few tables to help the reader with
intuitions.